# Facemasks: A Looming Microplastic Crisis

**DOI:** 10.3390/ijerph18137068

**Published:** 2021-07-01

**Authors:** Janith Dissanayake, Cecilia Torres-Quiroz, Jyoti Mahato, Junboum Park

**Affiliations:** Department of Civil and Environmental Engineering, Seoul National University, 1 Gwanak-ro, Gwanak-gu, Seoul 08826, Korea; janith1993@snu.ac.kr (J.D.); ctorresq@snu.ac.kr (C.T.-Q.); jyotimahato@snu.ac.kr (J.M.)

**Keywords:** microplastics, facemasks, COVID-19, solid waste management, water pollution

## Abstract

Single-use disposable facemasks have been used as a preventive measure against the ongoing COVID-19 pandemic. However, many researchers have found evidence that these facemasks are being dumped into lakes, rivers, and open garbage dumps. Facemasks have the potential of releasing microplastic fibers into the environment; a phenomenon that has been poorly investigated. Moreover, microplastic fibers composed of plastics have the potential of affecting the flora and fauna of many ecosystems. In this preliminary study, we investigate how many microplastic fibers had been released to the water by KF-AD, KF94, surgical, and FFP1 standard facemasks, which are the most widely available facemask standards in South Korea. The waterbody in our research was mechanically agitated for 24, 48, and 72 h. Findings showed that most of the layers of facemasks are composed of polypropylene. The surgical and KF94 standard facemasks released the highest number of microplastic fibers. Furthermore, under our research conditions, a single facemask can release at least 47 microplastic fibers per day (e.g., KF-AD standard mask), which can lead to the release of at least 1381 million microplastic fibers per day in total in South Korea if 70% of the urban population uses a single mask every day. Moreover, the released microplastic fibers significantly increased when the agitation time extended from 24 to 48 h. This finding suggests that the number of released microplastic fibers is likely to increase drastically.

## 1. Introduction

With the onset of the COVID-19 (SARS-CoV-2) pandemic, plastic demand attributable to medical waste has risen by 370%, while the packaging plastic demand has increased by 40% [1]. As facemasks and hand hygiene are considered to be the most effective non-pharmaceutical interventions [2] in preventing the spread of COVID-19, we can expect a continued use of facemasks despite the rollout of vaccination programs. However, most of the facemasks have not been disposed properly, and most of them have been dumped in open spaces. Consequently, facemasks have found their way to soil matrices and inland water bodies, and ultimately, to the sea. De-la-Torre and Aragaw [3] and Aragaw [4] recently reported that facemasks contain polypropylene based on their Fourier transformation spectrometric analysis. Furthermore, they suggested that polypropylene fibers may be released to the environment due to the mechanical stresses induced by water bodies. Tareq et al. [5] argued that 3D-printed facemasks may solve the solid waste crisis associated with the single use facemasks because most of the 3D-printed facemasks were made from thermoplastic polymers that can be recycled. However, after several recycling cycles, these 3D printed masks will become solid waste that can release microplastic fibers to the environment. Microplastic fibers can penetrate food chains and harbor the growth of microorganisms [6,7]. Therefore, the risk of microplastic fibers released from facemasks should be quantified to alert the relevant authorities in devising the necessary responses. In this regard, Saliu et al. [8] investigated the weathering effect of microplastic fibers released from facemasks in marine environments. They reported a substantial release (135,000 fibers on the average) of microplastic fibers to saline water. In their study, surgical facemasks were subjected to artificial weathering. However, their study did not investigate whether the number of released fibers would vary with the type of facemask (i.e., the filter standard of the facemask).

Here, we provide the first research involving the preliminary quantification of the number of microplastic fibers released by different facemasks to an aqueous medium.

## 2. Methodology

### 2.1. Identification Quantification of Released Microplastic Fibers

In South Korea, the most common types of facemasks are the surgical, KF-AD, and KF94 facemasks. These facemasks are the nonwoven types and designed with elastic ear loops and a nose strip. The filter efficiency of KF94 standard masks is higher (>95%) compared with the KF-AD standard and surgical masks, and its price is also higher. For the purpose of this experiment, we purchased three KF94, KF-AD, and surgical masks (i.e., three brands, with four masks from each brand) that are commonly available at convenience stores across the country. Furthermore, we purchased four FFP1 type facemasks (with CE certification) produced in the United Kingdom for comparison.

First, we cut one mask of each type along the border that sealed all the layers together and separated the layers. Further, the metal nose strip and the elastic ear loops were removed. The layers were individually analyzed for their chemical structure via Fourier transformation infrared (FTIR) spectroscopy (model: Nicolet 6700, Thermo-Scientific, Waltham, MA, USA). Then, the remaining masks were placed into 250 mL glass flasks. One flask contained only one facemask before the experiment commenced. We added 200 mL of MiliQ water (Aqua Max TM–Ultra Basic 360 series and Ultra 370 series, Youngin Chromass, Dongan-gu, Anyang, Gyeonggi-do, South Korea) to the flasks, and then the flasks were placed on a mechanical orbital shaking bench (OS-400, Hanyang Scientific Equipment Co. Ltd., Seoul, South Korea). The shaking speed was set to 150 rpm, and the system was mechanically shaken for 24, 48, and 72 h. Saliu, Veronelli, Raguso, Barana, Galli and Lasagni [8] used a higher rpm for mechanical shaking because the turbulences created by wave action in coastal waters were simulated. By contrast, our study considered a lower rpm rate to simulate the less turbulent water bodies. For all of the shaking times considered in this work, we triplicated the blank samples to measure the background contamination.

After mechanical shaking, the mask was gently extracted from the flask, and the MiliQ water solution was filtered through a Whatman nitrocellulose filter (Ø = 4.5 cm and pore size = 0.45 µm). A glass filtration apparatus attached to a vacuum pump was used to filter the water solution. Then, the filters were placed into an aluminum case to prevent contamination by airborne particles and left to dry for 24 h. Subsequently, the filters were observed using a Dino-Lite Edge digital microscope (Amno Electronics Corporation, Taipei, Taiwan) to confirm the homogeneous distribution of the fibers. The total number of fibers were derived according to the method provided by De Falco et al. [9] and Saliu, Veronelli, Raguso, Barana, Galli and Lasagni [8].

### 2.2. Calculation of Total Number of Released Microplastic Fibers

The number of masks (per day) to be used by the South Korean population was calculated on the basis of the equation suggested by Nzediegwu and Chang [10]. In their work, the daily mask usage is calculated as follows: total population (P) × percentage of urban population (Ɣ) × acceptance rate of mask usage (δ) × daily number of masked used per day per person (β). In Korea, the government has started to require citizens to wear facemasks in public spaces. Thus, we considered the δ values to be 0.7, 0.8, and 0.9. As the number of masks worn by a person in a day would vary depending on various factors, we considered four β values ranging from 1 to 4. The population of Korea is 51,307,107, among which the urban population is 81.8% as of 5 May 2021 [11].

## 3. Results

### 3.1. FTIR Spectral Analysis

The chemical compositions of all layers of facemasks considered in this study are illustrated in Appendix A, and a corresponding summary is presented in Figure 1. Among the 13 investigated layers, 11 of them (>84%) demonstrated band patterns similar to that of polypropylene. The innermost layer of the FFP1 facemask and the second layer of the KF94 mask were the only exceptions. The band patterns of these two facemask types showed high similarities to the band pattern of polystyrene. Similar observations were established for PS/nonwoven, pure polystyrene, and nonwoven fabrics by Diao et al. [12], Zhang et al. [13], and Aragaw [4]. Besides, Potluri and Needham [14] and Liebsch [15] reported that a single N95 mask and a disposable surgical facemask contain approximately 11 g and 4.5 g of polypropylene and other plastic derivatives, respectively. Under certain environmental conditions, such as exposure to UV radiation, high pH, or high-temperature environments, the plastic components may disintegrate, eventually creating micro/nanoplastics. As polypropylene is highly persistent, the complete degradation of these microplastic fibers is almost impossible.

### 3.2. Microplastic Fiber Count

The sizes of the fibers retained by the filter were observed using a Dino-Lite Edge Digital Microscope (Amno electronics corporation, Taipei, Taiwan) (Figure 2). The size of the fibers released by the facemasks fit the definition of microplastics (<3 mm) with some fibers having a size less than 1 mm. Furthermore, as illustrated by the microscope images before and after shaking in the water, the gaps between fibers have widened, possibly as a result of fiber leaching. The microplastic fibers released by all facemask types after 48 h are shown in Figure 3. The experimental results showed that, after 48 h of mechanical shaking, a considerable number of fibers are released to the water by all of the facemask types. The highest number of fibers (202) were released by surgical masks, followed by KF94 (161) and FFP1 (160) types. The KF-AD type released the least number of fibers (74). The mean number of released fibers were 81, 147, 169, and 143 for the KF-AD, KF94, surgical, and FFP1 facemasks, respectively. The corresponding standard deviations were 7, 18, 31, and 16. These values were significantly higher than those obtained in the triplicated blank runs, which were conducted to determine the background contamination. For the blank runs, we observed a mean of 2 microplastic fibers retained on the filter. This finding may be attributed to the extensive use of facemasks by laboratory users. The mean masses of facemasks were 1.289, 2.303, 1.179 and 4.004 g for KF AD, KF 94, surgical and FFP 1 facemasks, respectively. The surgical masks released a higher number of fibers even though they had the least mass. This could be due to number of reasons such as the manufacturing method. Furthermore, the porosity of the surgical masks is higher compared to others (with less filtration efficiency) thus, the flow through these pores may have resulted in higher fragmentation and release of fibers to the water.

### 3.3. Variation of Number of Released Microplastic Fibers over Time

When the shaking time increased, the number of released microplastic fibers drastically increased (Figure 4). The number of fibers released by the KF-AD, surgical, and FFP1 facemasks increased by a factor greater than 2.5, whereas the fibers released by the KF94 type increased by a factor of 2 when the shaking time increased from 24 h to 48 h. The number of fibers released by the surgical masks after 48 h of shaking corroborated the findings of Saliu, Veronelli, Raguso, Barana, Galli and Lasagni [8] who investigated the effect of weathering on the release of surgical mask microplastic fibers in saline waters. In their study, the surgical masks released an average of 398 fibers to artificial seawater without the effect of weathering. When the facemasks were subjected to artificial weathering, the masked released 135,000 fibers on the average, resembling a 0.2% loss of mass. Here, we did not consider the weathering effect; thus, the number of fibers released by the masks when exposed to UV radiation and high temperatures may be significantly higher. A more detailed presentation of the data in Figure 3, with standard deviations, is shown in Appendix A. For all of the mechanical shaking times considered in this study, the surgical masks were found to release the highest number of microplastic fibers, followed by the FFP1 type.

### 3.4. Impact on Tertiary Environments

Then, the total number of used masks were calculated, as previously described in the Methodology section. We calculated the total number of microplastic fibers that can be released to the environment by using the number of microplastics released by a KF-AD facemask after 24 h of mechanical shaking (Table 1). Even with the lower acceptance rates and daily facemask usage, at least 1381 million microplastic fibers can be released to the environment every day. In Korea, the majority of the masks are treated as a general waste and sent to be landfilled [16]. With the infiltration of rainwater, the fibers can be transported with leachate. However, as emphasized by Fadare and Okoffo [17], De-la-Torre and Aragaw [3], and Gorrasi, Sorrentino and Lichtfouse [1], facemasks also find their way to aquatic and soil ecosystems due to poor solid waste management practices. The gravity of this situation with respect to the devising of proper disposal policies should be urgently explored.

## 4. Discussion

The aim of this study is to quantify the number of microplastic fibers released by different types of facemasks that are being used in South Korea. Although the standards applicable to facemask manufacturing vary among countries, the compositions of the facemasks are similar.

The findings of this research can be used to prompt governmental and non-governmental actions about the proper disposal of facemasks. Notably, although the situation is generally alarming, the number of released microplastic fibers presented in this study is conservative because the weathering, abrasion, and turbulence conditions in aquatic environments have not been considered. In other words, the number of fragmented microplastic fibers released by facemasks to aquatic bodies are likely significantly higher than our findings. With the current projections depicting 2.4–52 billion facemasks produced in 2020 [18,19], an estimate of 72–31,200 tons of microplastics may be released to the oceans in the same year [8].

The negative implications of microplastics have been reported by many researchers [20,21,22,23]. Microplastics are carriers of pathogenic microorganisms and develop biofilms on their surface [24]. These situations can prolong the survival of pathogenic microorganisms in the environment, especially during the COVID-19 [25] pandemic, and facilitate the recurrence of outbreaks. Furthermore, the biomagnification of microplastics by aquatic food chains [26] may compromise food security if facemasks are not properly disposed. In countries where no proper waste disposal methods are practiced, the fibers can penetrate the soil matrix as a consequence of open dumping. The fibers can infiltrate the soil column and may even reach the ground water level. Not much research is available on the impact of microplastics on subsoil layers. Horton et al. [27] recommended to investigate the impact of microplastics on terrestrial environments, as bioaccumulation and co-transported chemicals may cause direct and indirect hazards in terrestrial species, especially since some of those species have parallel taxonomies to aquatic species. Moreover, nanoscale plastics in soils can alter the soil biota, and the application of sewage sludge in agriculture may further result in microplastic pollution [28]. Therefore, more research needs to be conducted to quantify nanoscale microplastics and investigate their impact on the subsoil environment.

Many researchers have investigated the possibility of reusing facemasks after their disinfection with UV radiation. Choi et al. [29] have recently developed a biodegradable facemask from chitosan. However, these technologies remain unpopular and unknown. Therefore, investing in suitable technologies may alleviate the severity of the looming microplastic crisis.

## 5. Conclusions

Single-use facemasks are being used globally in massive quantities since the onset of COVID-19. However, as demonstrated in this preliminary study, almost all types of facemasks are composed of polypropylene and release microplastic fibers to the environment on a daily basis. Our findings indicate that surgical facemasks release the highest number of microplastic fibers (>100), followed by the FFP-1 and KF-94 standard types, after mechanically shaking them in a water column. Furthermore, we can expect much more fibers to be released when facemasks are deposited in water bodies for a prolonged time due to weathering. The total amount of released microplastic fibers in South Korea can exceed 1381 million microplastic fibers per day. This number is even expected to rise exponentially if the number of facemasks used by a person within a day (β) also increases. The methodology used in this study can be adopted by researchers in other countries to quantify the microplastic fibers released by facemasks. The impact of these fibers on different ecosystems has still not been extensively studied. Proper solid waste management policies should be implemented at a global scale to prevent facemasks from reaching aquatic or soil ecosystems. We did not investigate the effect of water quality parameters on the release of these fibers; thus, we recommend further research to be conducted to understand the dynamics associated with the release of microplastic fibers in different aquatic environments. In addition, further research is recommended to determine the impact of these microplastic fibers and develop new technologies that can overcome the issues related to the release of microplastic fibers from single-use facemasks.

## Figures and Tables

**Figure 1 ijerph-18-07068-f001:**
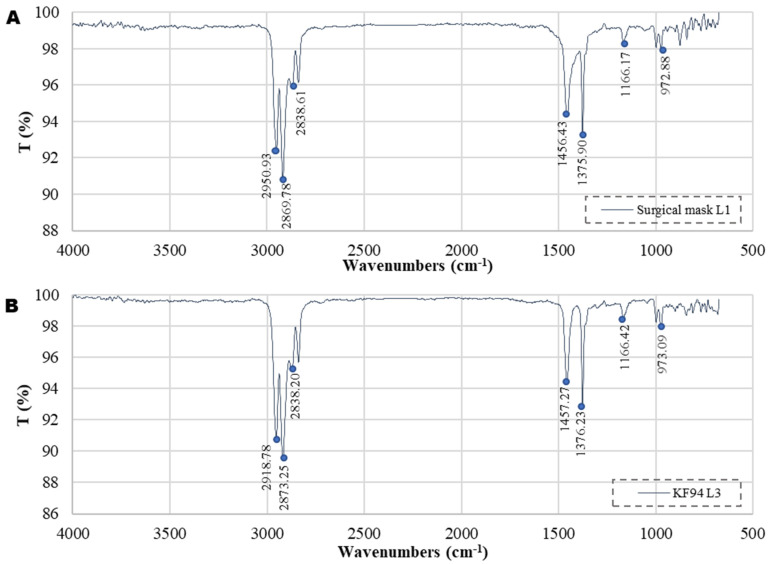
FTIR spectra of (**A**) outer layer of surgical mask (L1), (**B**) third inner layer of KF94 facemask (L3), (**C**) second layer of KF-AD facemask (L2), and (**D**) innermost layer of FFP1 mask (L3).

**Figure 2 ijerph-18-07068-f002:**
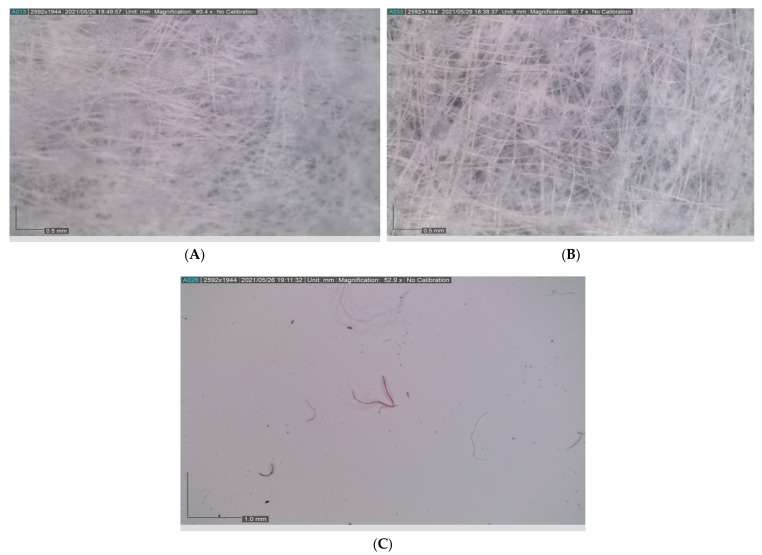
Fibers of the inner layer of a surgical facemask observed by a stereomicroscope (**A**) before shaking in water, (**B**) after shaking in water, and (**C**) with microplastic fibers retained in the filter.

**Figure 3 ijerph-18-07068-f003:**
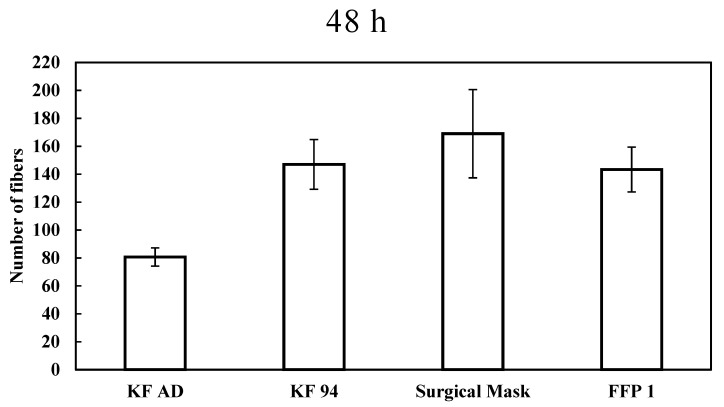
Number of microplastic fibers retained on the filter after shaking in MiliQ water at 150 rpm for 48 h.

**Figure 4 ijerph-18-07068-f004:**
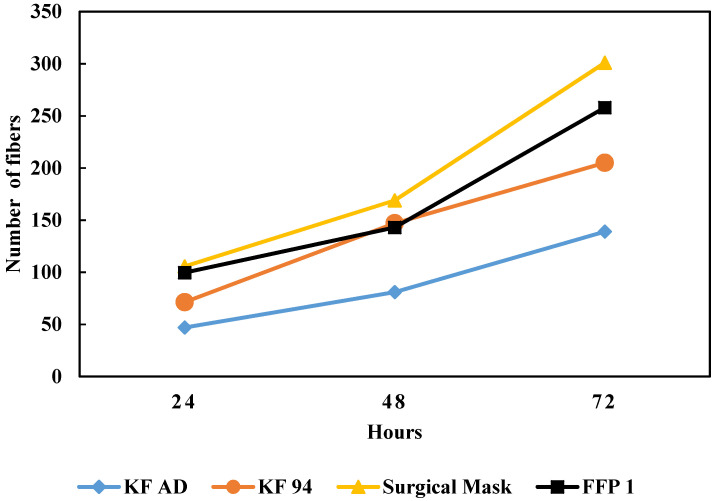
Number of microplastic fibers released to the water by KF-AD, KF94, surgical, and FFP1 facemasks after 24, 48, and 72 h of mechanical shaking.

**Table 1 ijerph-18-07068-t001:** Estimated number of facemask usage per day and number of released fibers versus various acceptance rates and number of facemasks used per person per day.

β	Number of Masks (Millions)			Number of Fibers Leached 24 h (Millions)		
	δ			δ		
	0.7	0.8	0.9	0.7	0.8	0.9
1	29	34	38	1381	1578	1775
2	59	67	76	2762	2536	2853
3	88	101	113	4142	3805	4280
4	118	134	151	5523	5073	5707

## Data Availability

Data can be made available upon request.

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
