# Peer review of "Facemasks: A Looming Microplastic Crisis"

_ijerph, 2021, doi:10.3390/ijerph18137068_

Round 1
Reviewer 1 Report
In this manuscript authors conducted an indepth experimental study to understand the burden of microfibers coming out of different types of facemasks used for the COVID-19 protection. For this, they selected four different types of facemasks dominantly used in south korean perspective. The manuscript is well organised and well-written. The study is important and timely. Hence, I would recommend it for publication after minor revisions. In this case, following points are to be clarified/added:
- As per Fig 3, it is clear that the biggest amount of microfibers are from Surgical mask. The reasons must be explained in the relevant section. Moreover, the amount of microfiber released is related to the weight/size of a facemask?
- Before Discussion section (Section 4), it is always mentioned about 'microfiber', however, in Section 4, 'microplastic' has been used (even microplastic fiber is used). I would suggest to use consistent word from the beginning
- Authors nicely mentioned about biodegradable facemasks etc at the end (Conclusion). However, I would like to mention about 3D printed facemasks at the Introduction section. For various 3D printed facemasks (some of them are biodegradable too), please mention this reference, https://www.sciencedirect.com/science/article/pii/S0278612520302351
Finally, I must say it is a good manuscript to be accepted in this journal after these minor modifications
Author Response
Please find our response attached to this submission.

Reviewer 2 Report
The Authors proposed an interesting topic for research. The abstract contains relevant information about the scope of research and results obtained. Methods are described with sufficient details. The paper is well organized and the data are well presented. Only the bibliography should be standardized according to the Journal. I recommend to publish this paper.
Author Response

(The authors gave the same response as above.)

Reviewer 3 Report
This is a good paper in that (1) some novel findings were presented in the context of differential microplastic fibers released from different brands of facemasks; (2)The surgical and KF94 standard facemasks released the highest number of microfibers in the described conditions;
However, this paper still has apparent drawbacks as follow:
1)the absolute number itself of 1381 million microfibers per day itself may have no meaning, unless the size range of microfibers and some detailed harm are provided or compared;
2)How to cut a mask in detail is not clearly documented. Should clearly stated that each mask in one flask or four masks in one flask?
3)Besides MiliQ water, filtered natural water, including sea water, river water, tap water and the ground water in garbage dumping place, shall be tested and compared. Or try water with different pH and salinity;
4)The author shall clearly describe if the fibers already exist in facemasks before dumped; if already exist, whether they would directly release onto human face skin (the small humid contact between facemask and human face skin can trigger the release/);
Hopefully the authors will revise the manuscript and even supplement with extra experiments to get the research much better.
Author Response

(The authors gave the same response as above.)
